# Longitudinal Changes of the Ruminal Microbiota in Angus Beef Steers

**DOI:** 10.3390/ani12091066

**Published:** 2022-04-20

**Authors:** Jeferson M. Lourenco, Taylor R. Krause, Christina B. Welch, Todd R. Callaway, T. Dean Pringle

**Affiliations:** Department of Animal and Dairy Science, University of Georgia, Athens, GA 30602, USA; jefao@uga.edu (J.M.L.); taylor.krause@uga.edu (T.R.K.); christina.welch@uga.edu (C.B.W.); dpringle@uga.edu (T.D.P.)

**Keywords:** calf, cow, dam, metabolic pathway, microbiome, rumen development

## Abstract

**Simple Summary:**

The rumen is a crucial organ in the digestion process of bovines; however, in beef cattle, it is not fully developed until sometime after weaning. In the same way, the microbial population that inhabits the rumen is constantly changing as animals age. The initial inoculation of the rumen of beef calves is heavily influenced by the environment, including the presence of adult animals in the pasture such as the cows. This study investigated the longitudinal changes that occur in the ruminal microbiota of Angus beef steers from weaning to slaughter. Ruminal samples were collected from 12 cows and their steer calves on weaning day, followed by subsequent collections on the same group of steers as they entered the feedlot, upon leaving the feedlot, and at the slaughterhouse. Results revealed that the ruminal microbial composition of the steers at a younger age (at weaning) was very similar to the ruminal microbiota of the adult cows; more so than it was from their own microbiota at later ages.

**Abstract:**

The ruminal microbiota of Angus cows and steers were characterized using 16s rRNA gene sequencing, and the expression of their metabolic pathways was predicted. Samples were collected on weaning day from the steers and the cows, and subsequently on three other occasions from the steers. Results showed that microbial richness, evenness, and diversity decreased (*p* < 0.001) in the rumen of the steers as they were weaned and transitioned to a high-concentrate feedlot diet. However, on the day of weaning, microbial evenness was similar to that observed in the rumen of cows (*p* = 0.12). The abundance of archaea was similar (*p* = 0.59) between the cows and steers at weaning, but it decreased (*p* = 0.04) in the rumen of steers after weaning, and remained stable (*p* ≥ 0.44) for the remainder of their lives. Likewise, no difference (*p* = 0.51) in the abundance of Bacteroidetes was detected between the cows and the calves on the day they were weaned, but the abundance of this phylum increased (*p* = 0.001) and remained stable after that. These results suggest that cows may have a strong influence on the composition, and help modulate the ruminal microbiota of young calves; however, following weaning, their ruminal microbiotas tend to differentiate from that state observed at earlier ages.

## 1. Introduction

The microbial population of the rumen is crucial to the ability of the animal to degrade cellulosic forage and allows the ruminant animal to thrive in environments that monogastric animals cannot [1]. Ruminant animals are born with a nearly sterile gastrointestinal tract that quickly becomes colonized by a succession of microbes during and after birth due to passage through the birth canal and subsequent maternal contact [2,3,4]. Immediately following birth, calves are constantly exposed to the maternal gastrointestinal microbial population via the udder, milk, and maternal licking [2,4]. However, maternal contact does not appear to be a persistent microbial seeding route for the gastrointestinal tract of calves [5]. This relationship underscores the fact that the dam is a significant driving force of inoculation and colonization of the rumen of beef calves, but the length of this impact is unclear. Furthermore, the temporal extent of maternal influence on the calf gastrointestinal microbial population remains unknown [5].

The lack of a microbial population at birth means that young calves are not functionally ruminant animals [6]. However, as the ruminal fermentation begins producing short-chain fatty acids (SCFA), it causes the rumen epithelial tissue to grow faster than the rest of the gastrointestinal tract (allometrically) [4,7]. The composition of the ruminal microbial population determines the end products of the ruminal fermentation, and therefore the amount of SCFA available to the ruminant that affects the subsequent rate of ruminal development [8,9].

Beef calves typically remain with their dams until weaning at 6–8 months of age. Following weaning, calves are dependent on the ruminal microbial degradation of feedstuff for all their nutritional requirements. As calves mature, they often are reared under several different production scenarios (e.g., backgrounding, feedlot) where they consume very different rations ranging from pasture to high grain rations in the feedlot. The ruminal microbial population is plastic and changes in response to different feedstuffs utilized in each production phase, yet it is unclear how the early ruminal microbial composition of calves is predictive of the microbial population of the adult animal. Therefore, the present study was designed to evaluate the changes in the ruminal microbiota of Angus beef steers from weaning until slaughter. Ruminal samples from cows and their steer calves were collected at weaning, and ruminal microbial populations were again evaluated at feedlot entry and departure as well as arrival at the abattoir.

## 2. Materials and Methods

### 2.1. The Animals Used in the Study

All animal procedures performed in this study were reviewed and approved by the University of Georgia’s Animal Care and Use Committee (AUP #A2012 11-006-R1).

Angus cattle selected for six generations based on predicted carcass intramuscular fat (marbling) and feed conversion (residual average daily gain) were used in the current study. The selection criteria and purpose are described elsewhere [10,11], and are unrelated to the purpose of the current study. In brief, the steers were produced from a cow herd located at the University of Georgia’s research and education center located in Calhoun, GA (34°30′ N, 84°57′ W). They were born in the spring of 2018 and reared in a pasture-based system until weaning at approximately 7.5 months of age. Following weaning, the steers were backgrounded on pasture until they entered the feedlot-finishing phase at approximately 13 months of age. The feedlot was located in Brasstown, NC (35°10′ N, 83°23′ W) and the animals stayed there for approximately 4 months before being slaughtered at the UGA Meat Science Technology Center, in Athens, GA (33°57′ N, 83°22′ W). The composition of the diet fed during the feedlot-finishing period can be found in Appendix A.

### 2.2. Collection and Processing of Samples

During the described life cycle, ruminal samples were collected from the steers at different timepoints, namely: (1) On their weaning day; (2) When they started receiving a high-grain diet in the feedlot; (3) During their last week in the feedlot; (4) Upon arriving at the slaughterhouse (lairage). Additionally, on weaning day, ruminal samples were also collected from their mothers. All samples were collected by esophageal tubing as previously described [12,13]. Briefly, this method uses a hose with a perforated probe, which was introduced in the animal’s forestomach, and through the use of a vacuum pump, approximately 300 mL of ruminal fluid was obtained from each animal. A subsample of approximately 45 mL was then transferred to a sterile conical tube and immediately placed on ice. The samples were then transported to the laboratory and stored at −80 °C until further processing. Once all samples had been gathered, total DNA extraction was performed using a methodology previously described [14], except that the DNA purification and elution processes were performed manually instead of using a robotic workstation. The procedure utilized for DNA extraction used a combination of mechanical and enzymatic methods to obtain the genomic DNA. Briefly, 0.33 g of sample was placed into a Lysing Matrix E Tube (MP Biomedicals, Solon, OH, USA), which was homogenized in a FastPrep 24 homogenizer (MP Biomedicals, LLC, Irvine, CA, USA) at 6.0 m/s for 40 s. Following, the samples were further processed using a QIAamp Fast DNA Stool Mini Kit (Qiagen Inc., Germantown, MD, USA) following the manufacturer’s instructions. The purified DNA was eluted in a total volume of 100 µL. The DNA concentrations were determined spectrophotometrically using the Synergy H4 Hybrid Microplate Reader (BioTek Instruments, Inc., Winooski, VT, USA). Samples with a minimum concentration of 10 ng/µL of DNA were stored at −20 °C until further analysis. Samples that failed to meet that requirement were rejected and subjected to a new DNA extraction cycle.

### 2.3. DNA Sequencing and Bioinformatics

Microbial DNA sequencing was performed at the Georgia Genomics and Bioinformatics Core. Library preparation was performed using the S-D-Bact-0341-b-S-17 (5′- CCTACGGGNGGCWGCAG-3′) forward and S-D-Bact-0785-a-A-21 (5′-GACTACHVGGGTATCTAATCC-3′) reverse primers. This primer pair amplifies the V3-V4 hypervariable regions of the 16S rRNA gene, producing an expected amplicon size of 464bp, and can capture up to 64.6% of the domain *Archaea* [15]. Each PCR reaction contained a DNA template (12.5 ng), 5 μL forward primer (1 μM), 5 μL reverse primer (1 μM), 12.5 μL 2× Kapa HiFi Hotstart ready mix (Roche Molecular Systems, Inc., Indianapolis, IN, USA), and water to a final volume of 25 μL. The DNA was subjected to initial denaturation at 95 °C for 3 min. Amplification was then achieved by 25 cycles of denaturation at 95 °C for 30 s, annealing at 55 °C for 30 s, and extension at 72 °C for 30 s. Final extension was at 72 °C for 5 min. PCR products were cleaned using AMPure XP magnetic beads (Beckman Coulter, Inc., Indianapolis, IN, USA) and 80% ethanol. The PCR products were then submitted to another round of PCR to incorporate indexes to the samples (Illumina Nextera XT indexing primers, Illumina, Inc., San Diego, CA, USA). In this step, each PCR reaction contained 5 μL of each index primer, 25 μL 2× Kapa HiFi Hot Start Ready-mix (Roche Molecular Systems, Inc., Indianapolis, IN, USA), and 10 μL water. PCR cycling conditions were as previously described except for the number of amplification cycles, which was set to 8. PCR products were cleaned using AMPure XP beads and 80% ethanol, pooled, and paired ends were sequenced at a read length of 300 nucleotides on a MiSeq platform (Illumina, Inc., San Diego, CA, USA). A bacteriophage PhiX genome (PhiX Control v3 Library; Illumina Inc., San Diego, CA, USA) was used as a quality control for the sequencing runs.

Sequencing data were demultiplexed and converted into FASTQ files, and the paired-end sequences were converted into QIIME 2 artifacts. The non-biological nucleotides were removed, and sequences were denoised, dereplicated, and chimera-filtered using DADA2 [16]. Taxonomies were assigned to the sequences by using a pre-trained Naive Bayes classifier which was trained on the SILVA 138 SSU database [17], and reads were classified by taxon using the fitted classifier [18]. Samples were rarefied to 5041 sequences per sample prior to computing alpha and beta diversity using the “qiime diversity” plugin [19]. Phylogenetic Investigation of Communities by Reconstruction of Unobserved States (PICRUSt2) was carried out to make inferences about the metabolic functions of the microbial community [20]. Metagenome metabolic functions were assessed using the MetaCyc pathway database [21]. Expression of the pathways in the rumen of the cows was set to 100%, and the expression observed in the other samples (steers at the different stages of their life cycles) was calculated relative to the ones observed in the cows.

### 2.4. Data Availability and Statistical Analysis

Nucleotide sequencing data were deposited in a public repository (MG-RAST) [22] under accession number mgm4944536.3. Statistical analyses were performed using the software R (v3.3.3; R Foundation for Statistical Computing, Vienna, Austria), and Minitab (v18.1; Minitab, LLC State College, Pennsylvania, USA). Independent ANOVAs were performed between all timepoints (e.g., at weaning vs. beginning feedlot; beginning vs. end of feedlot; etc.) using each animal as an experimental unit. In addition, data from the cows were compared to the steers at weaning. Results were declared statistically significant when *p* ≤ 0.05.

## 3. Results

### 3.1. Microbial Richness, Evenness, and Diversity

Beta-diversity analysis (Figure 1) revealed two major clusters: one containing the ruminal samples of cows and the steers at weaning, and another cluster comprising the samples taken from the steers at the beginning and end of the feedlot phase, as well as upon arrival at the slaughterhouse (lairage). The three main features driving the clustering of the groups were fadb51778536c78578228b3e6173ff16: Genus Prevotella (uncultured_rumen); fb044135d491429c8c4f69dcbab3ec63: Genus Prevotella (uncultured_Prevotella); and 580de5d0d4c9377e0154cd1187efb7c1: Genus F082 (uncultured_rumen). The number of features detected in each sample type, and sample evenness (Figure 2) were numerically greater for the cows, followed by steers at weaning, and had a very significant drop (*p* < 0.001; Table 1) when steers entered the feedlot, and remained lower for the other phases. Similarly, microbial diversity (Figure 3), expressed both as Shannon diversity index and Faith’s phylogenetic diversity index had a similar behavior, with higher values observed for the adult cows, followed by the steers on weaning day.

### 3.2. Specific Microbial Taxa

Overall, 99.85% of our sequences were classified at the phylum level, 99.5% were classified at the family level, 97.5% at the genus level, and less than 50% could be identified at the species level. The archaeal population was greater in the animals that were grazing, i.e., the cows and their weanling steers (Figure 4); however, a sharp decrease (*p* = 0.04) in archaea was detected when comparing the steers at weaning to the samples obtained when they entered the feedlot. After this transition to a high-grain feedlot diet, archaeal populations remained lower than what was observed at weaning. The average abundance of archaea in all samples was 0.28%. When investigating at deeper taxonomic levels, we found that the predominant archaeal taxon was *Methanobrevibacter*, with an average abundance of 0.27%.

At the phylum level, Figure 5 shows the great degree of similarity between the ruminal microbiota of the steers on weaning day to the rumen population of the cows. However, following weaning, the abundance of Bacteroidetes increased (*p* = 0.001) and remained higher than at the earlier stages (*p* ≥ 0.22). Conversely, the ruminal abundance of Verrucomicrobia and Planctomycetes was numerically greater in the cows and steers at weaning, but dropped (*p* ≤ 0.001) after steers were weaned. The abundance of the other major phylum—Firmicutes—fluctuated from 21.6 to 29.9% in all sample types and the contrasts were not statistically significant (*p* ≥ 0.10).

### 3.3. Expressed Metabolic Pathways

The predicted expression of some relevant microbial metabolic pathways (Table 2) is in alignment with the microbiota results, as they reveal an overall similarity between samples collected from the cows to the ones collected from the steers at weaning, the only exception being for L-histidine degradation I, which was greater in the cows (*p* = 0.03). Interestingly, the expression of all the metabolic pathways shown in Table 2 significantly changed (*p* ≤ 0.03) when comparing the steers at the beginning of the feedlot phase to the samples obtained at weaning. Moreover, there were no significant changes when comparing the ruminal samples collected at the beginning to the ones from the end of the feedlot phase (*p* ≥ 0.20). Likewise, no significant changes (*p* ≥ 0.06) were found when contrasting the ruminal samples collected at lairage to the ones obtained at the end of the feedlot phase.

## 4. Discussion

The similarity between the cows’ and the steers’ microbiota at weaning was evident from several different perspectives: from analyzing the overall microbial richness (number of observed features), microbial diversity (Shannon diversity and Faith’s phylogenetic diversity), microbial evenness, beta-diversity, and several individual microbial taxa. It is not clear, however, if the similarities between the cows and the weanling steers were a result of direct transmission of microbes from the cows to the calves, or if these two groups of animals developed similar microbiotas because they were on the same type of diet (pasture-based); or even if that was due to a combination of diet and direct transmission from the cows. In Holstein dairy cows, a comparison of the bacterial communities found in vaginal and fecal samples from 81 cows revealed that maternal vaginal microbiota potentially influences the initial bacterial colonization of the calf upper respiratory tract [23]. Moreover, a direct modulation of piglet microbiota through maternal microbial transfer has been suggested, and this modulation had some long-term effects on how pigs performed [24]. Since swine production occurs in a totally confined environment in which the conditions are more controlled, this indicates that the direct transfer of microbes plays an important role in the mother-offspring microbial similarity; and this was likely the case in the present study. Despite the maternal effect on the offspring’s microbiota, following weaning, calves are transitioned to a diet made exclusively of solid feeds. According to Clemmons and collaborators (2019), diet is one of the main drivers of microbial community composition and modulation in cattle [25]. Therefore, the differentiation observed in the calves’ microbiota following weaning in the present study is not unexpected.

The overall microbial richness, diversity, and evenness observed in the rumen of our animals were all greater for the cows. Besides being older (and consequently having their gastrointestinal tracts fully developed), the cows were exclusively on a forage-based diet, which is known to increase microbial diversity [26,27]. Interestingly, Pielou’s evenness index was similar for the cows and the steers at weaning, but as the steers were weaned and transitioned to a grain-based diet, ruminal evenness was reduced. Similarly, both microbial richness and diversity sharply decreased after the steers were weaned, and these traits tended to remain lower for the remainder of the steers’ productive life.

A recent study using cannulated Holstein cows [26] demonstrated the clear distinction between the rumen environment of cows fed a diet containing 70% forage compared to cows fed only 30% forage: The 70%-forage diet group maintained a greater ruminal pH throughout the day and promoted greater ruminal diversity and richness. In the present study, the second time point where the steers had samples collected was when they were in a feedlot, and therefore, on a high-grain diet. That was also the case for the subsequent time points. Consequently, the fact that their ruminal richness, diversity, and evenness never returned to the “at weaning” levels is in line with previous findings.

Although the primers used in the present study were not specifically designed for the domain *Archaea*, they are able to capture up to 64.6% of that domain [15]. Thus, even though not all of archaea are able to be represented in this study, this primer pair is still one of the best at showing the majority of the species within the domains of Archaea and Bacteria. The abundance of archaea in the rumen of cattle was similar (*p* = 0.59) for the cows and the steers on weaning day, but it dropped more than 50% after weaning and remained at similar levels for all the other time points. Differently from our results, Kumar and collaborators [28] did not see changes in archaeal diversity when dairy cows transitioned from a ration composed of 80% forages to one composed of 50% forage. However, Zhang and collaborators [29] reported a decreased number of archaeal species as the level of inclusion of concentrate in the ration increased, which is in line with what we observed.

With respect to the main phyla, the relative abundance of Bacteroidetes was 52.3 and 53.9% for the cows and steers at weaning, which was found to be similar; but following weaning, ruminal abundance of this phylum significantly increased to 64.7% and never dropped below 62% for all the other time points. In contrast, no significant differences were observed in the second most abundant phylum: Firmicutes. In line with our results, [30] also found an increase in Bacteroidetes as cows moved from a high-forage diet to a low-forage diet; however, they also observed a decrease in the population of Firmicutes, which we did not find.

Besides the changes observed in the microbiota, several differences were also observed in the expression of microbial metabolic pathways. Interestingly, virtually all the pathways that we studied were similarly expressed in the adult cows and the steers at weaning. As the steers progressed to the feedlot-finishing phase, the expression of those pathways markedly changed compared to the observations done at weaning; however, at the next timepoints (end of feedlot phase and slaughter), the expression of the pathways remained unchanged compared to the beginning of the feedlot. A good example of this is the pathway involved in methanogenesis from H_2_ and CO_2_, which was similarly expressed in the cows and the steers at weaning (*p* = 0.39) but decreased (*p* = 0.02) as animals moved into the feedlot, and remained steady (*p* ≥ 0.47) at the other two timepoints evaluated. Moreover, results from this methanogenic pathway are in line with the ones observed in the microbiota of our animals, which revealed ga reater abundance of methanogenic archaea at weaning compared to the later periods.

Contrarily to what was observed for methanogenesis from H_2_ and CO_2_, the expression of the gluconeogenic pathway using malate as a carbon substrate (i.e., gluconeogenesis I) increased as the steers transitioned to a high-grain diet, despite being similar to the expression of the cows on weaning day. This indicates that the generation of glucose by the microbiota starting from non-sugar substrates intensified during the feedlot, compared to the weaning stage when steers were on a forage-based diet.

## 5. Conclusions

This longitudinal analysis of the ruminal microbiota of Angus beef steers revealed that, at weaning, the microbial composition of the steers closely resembled the composition observed in the adult cows (i.e., their mothers). However, following weaning and transitioning to a feedlot-finishing setting, the microbiota of their rumens experienced significant changes, which included a decrease in the overall microbial richness and diversity, a decrease in archaeal population, and an increased abundance of Bacteroidetes. Furthermore, those changes persisted for the remainder of the steers’ productive life (assessed at the end of the feedlot phase and upon arrival at the slaughterhouse). The metabolic pathways expressed by their rumen microbiotas also changed and followed a similar pattern, with drastic changes following weaning. These findings suggest that the composition and function of the steers’ microbiota are heavily influenced by their mothers’, but change dramatically after weaning. Additionally, although the changes that occurred after weaning were likely due to the change in the steers’ diets (i.e., they moved from a pasture-based diet during the cow/calf phase to a high-grain diet in the feedlot), further studies are necessary to elucidate if the similarity of their microbiotas to the one of the cows at weaning is due to animal behavior/natural inoculation by the cows, or simply due to their diet.

## Figures and Tables

**Figure 1 animals-12-01066-f001:**
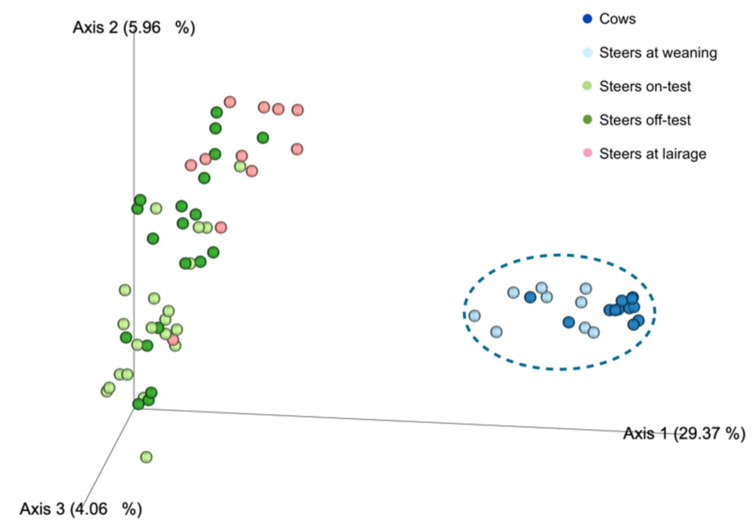
Principal coordinates analysis of Beta-diversity (Unweighted UniFrac) observed in the ruminal samples of the adult cows, steers on weaning day, steers at beginning of feedlot phase (on-test), steers at end of feedlot phase (off-test), and upon arriving at the slaughterhouse (lairage). PERMANOVA *p*-value = 0.001.

**Figure 2 animals-12-01066-f002:**
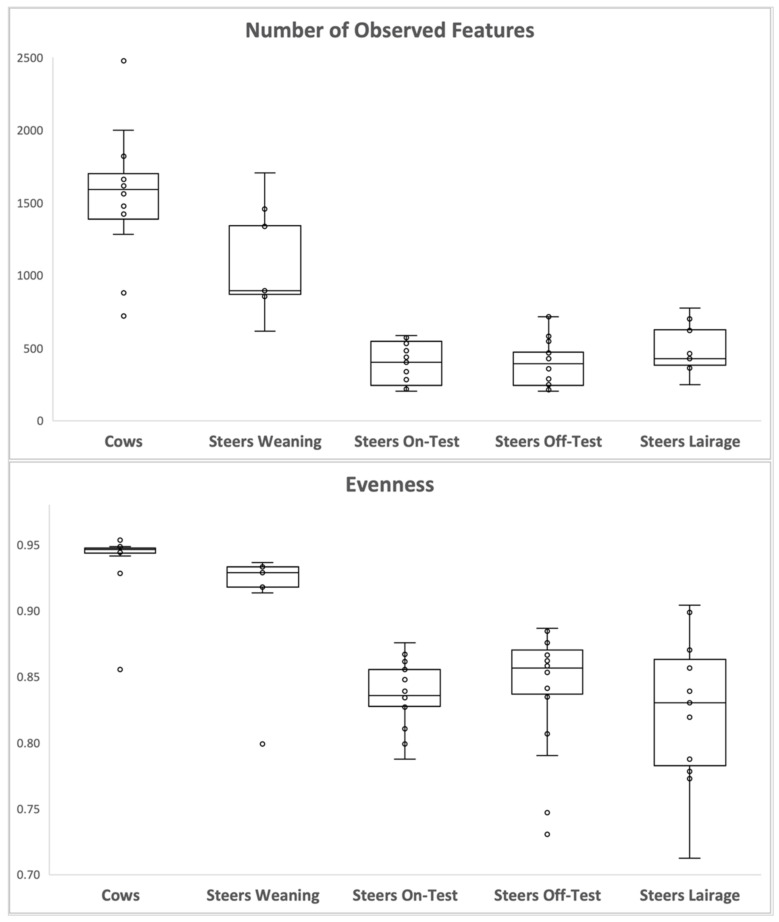
Number of observed features and Pielou’s evenness index observed in the ruminal samples of the adult cows, steers on weaning day, steers at beginning of feedlot phase (on-test), steers at end of feedlot phase (off-test), and upon arriving at the slaughterhouse (lairage). Average values and significance for the contrasts are presented in Table 1.

**Figure 3 animals-12-01066-f003:**
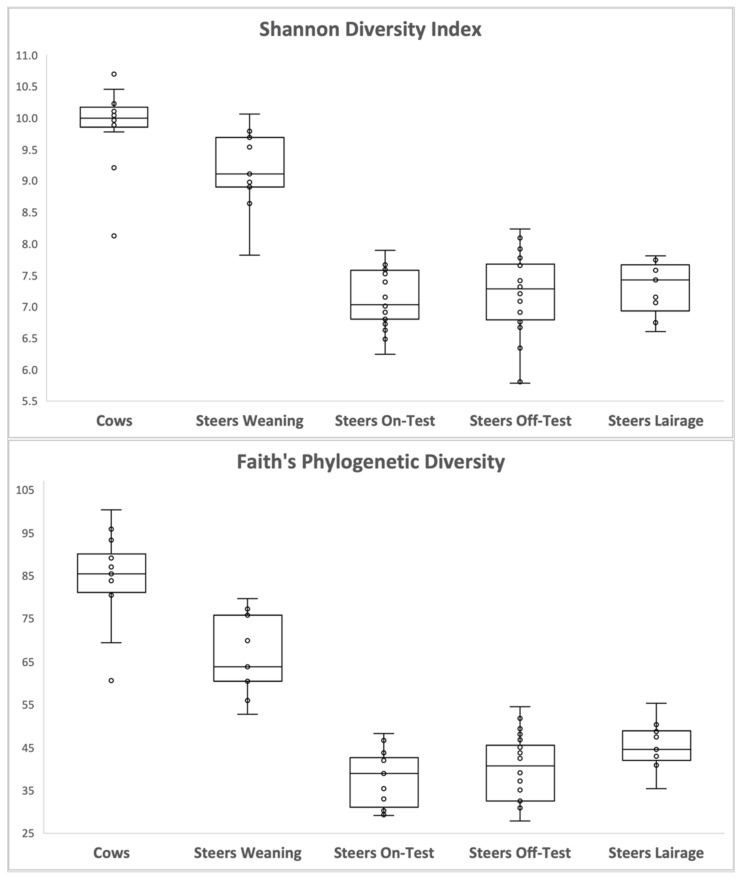
Shannon diversity index and Faith’s phylogenetic diversity index observed in the ruminal samples of the adult cows, steers on weaning day, steers at beginning of feedlot phase (on-test), steers at end of feedlot phase (off-test), and upon arriving at the slaughterhouse (lairage). Average values and significance for the contrasts are presented in Table 1.

**Figure 4 animals-12-01066-f004:**
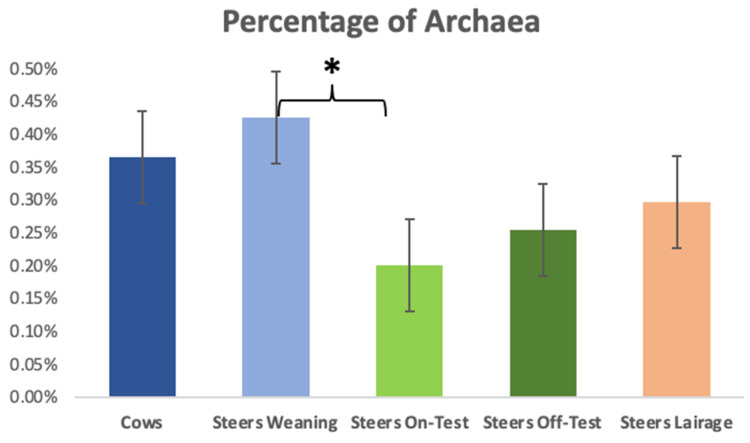
Percentage of archaea detected in the ruminal samples of the adult cows, steers on weaning day, steers at beginning of feedlot phase (on-test), steers at end of feedlot phase (off-test), and upon arriving at the slaughterhouse (lairage). A significant difference (*) was detected between steers at weaning and at beginning of the feedlot phase (*p* = 0.04); but not for the other contrasts (*p* ≥ 0.44).

**Figure 5 animals-12-01066-f005:**
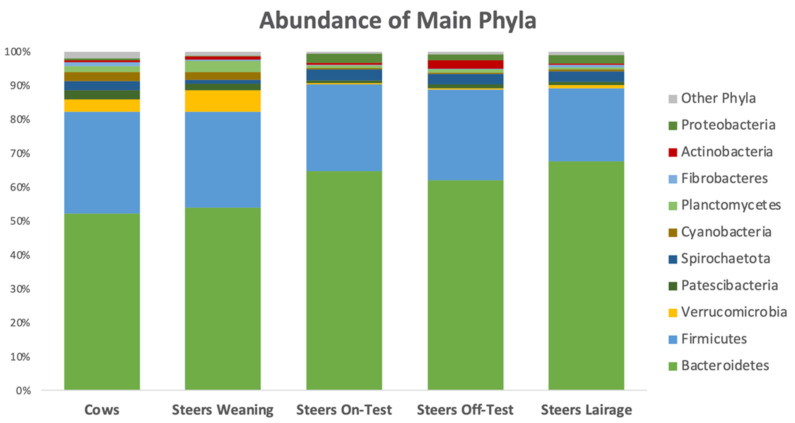
The abundance of the main phyla in the ruminal samples of the adult cows, steers on weaning day, steers at beginning of feedlot phase (on-test), steers at end of feedlot phase (off-test), and upon arriving at the slaughterhouse (lairage).

**Table 1 animals-12-01066-t001:** Number of observed features, Shannon diversity index, Faith’s phylogenetic diversity index, and species evenness observed in the ruminal samples of the adult cows, steers on weaning day, steers at beginning of feedlot phase (on-test), steers at end of feedlot phase (off-test), and upon arriving at the slaughterhouse (lairage).

Item	Cows	Steers Weaning	Steers On-Test	Steers Off-Test	Steers Lairage	Contrast 1	Contrast 2	Contrast 3	Contrast 4
Obs. Features	1545	1106	395	393	489	0.03	<0.001	0.96	0.12
Shannon Index	9.9	9.2	7.1	7.2	7.3	0.03	<0.001	0.83	0.64
Faith’s PD	84.3	66.3	37.8	40.1	45.4	0.001	<0.001	0.32	0.06
Evenness	0.937	0.912	0.837	0.843	0.824	0.12	<0.001	0.59	0.32

Contrast 1: *p*-value for the comparison cow vs. steers at weaning. Contrast 2: *p*-value for the comparison steers at weaning vs. steers at beginning of feedlot. Contrast 3: *p*-value for the comparison steers at beginning of feedlot vs. steers at the end of feedlot. Contrast 4: *p*-value for the comparison steers at end of feedlot vs. steers at the slaughterhouse (lairage).

**Table 2 animals-12-01066-t002:** Calculated expressions * of selected microbial metabolic pathways for the ruminal samples of the adult cows, steers on weaning day, steers at beginning of feedlot phase (on-test), steers at end of feedlot phase (off-test), and upon arriving at the slaughterhouse (lairage).

Item	Cows	Steers Weaning	Steers On-Test	Steers Off-Test	Steers Lairage	Contrast 1	Contrast 2	Contrast 3	Contrast 4
Arginine, ornithine, and proline interconversion	100%	117.1	68.8	76.1	49.8	0.13	<0.01	0.56	0.06
L-arginine biosynthesis I (via L-ornithine)	100%	99.9	89.9	88.8	87.8	0.92	<0.01	0.72	0.78
L-arginine biosynthesis II (acetyl cycle)	100%	97.6	82.4	83.1	79.5	0.30	<0.01	0.87	0.48
L-aspartate and L-asparagine biosynthesis	100%	102.6	123.6	120.1	113.4	0.40	<0.01	0.34	0.15
Pyruvate fermentation to butanoate	100%	93.7	55.5	62.5	76.0	0.51	<0.01	0.42	0.20
Tetrahydrofolate biosynthesis and salvage	100%	99.1	109.5	109.5	108.2	0.46	<0.01	0.98	0.70
Gluconeogenesis I	100%	101.1	110.0	110.3	108.6	0.40	<0.01	0.84	0.42
β-D-glucuronosides degradation	100%	101.0	125.7	113.9	106.9	0.89	0.03	0.20	0.51
L-histidine degradation I	100%	83.0	125.4	132.9	137.8	0.03	<0.01	0.42	0.70
Methanogenesis from H_2_ and CO_2_	100%	125.1	54.1	67.4	64.1	0.39	0.02	0.47	0.89
Methylerythritol phosphate pathway I	100%	101.8	108.5	108.3	104.8	0.16	<0.01	0.96	0.19
O-antigen building blocks biosynthesis	100%	99.9	85.0	90.3	82.4	0.98	<0.01	0.21	0.16
L-lysine fermentation to acetate and butanoate	100%	127.6	41.0	49.8	49.3	0.17	<0.01	0.36	0.97
Cob(II)yrinate a,c-diamide biosynthesis I	100%	103.9	43.8	46.7	68.9	0.60	<0.01	0.78	0.15
Methylerythritol phosphate pathway II	100%	101.8	108.5	108.3	104.8	0.16	<0.01	0.96	0.19
Phosphatidylglycerol biosynthesis I	100%	101.7	94.0	95.0	88.4	0.26	<0.01	0.71	0.10
Phosphatidylglycerol biosynthesis II	100%	101.7	94.0	95.0	88.4	0.26	<0.01	0.71	0.10
NAD salvage pathway I	100%	101.6	110.9	109.3	105.6	0.40	<0.01	0.56	0.26
UDP-N-acetyl-D-glucosamine biosynthesis I	100%	98.4	76.0	83.9	73.3	0.66	<0.01	0.24	0.23
**Average Expression**	**100%**	**103.1**	**89.8**	**92.2**	**89.4**	**-**	**-**	**-**	**-**

* Calculated expression for the steers was compared to the cows, which were set as the standard (100%). Contrast 1: *p*-value for the comparison cow vs. steers at weaning. Contrast 2: *p*-value for the comparison steers at weaning vs. steers at beginning of feedlot. Contrast 3: *p*-value for the comparison steers at beginning of feedlot vs. steers at the end of feedlot. Contrast 4: *p*-value for the comparison steers at end of feedlot vs. steers at the slaughterhouse (lairage).

## Data Availability

The nucleotide sequencing data used in this study were deposited in a public repository (MG-RAST) under accession number mgm4944536.3.

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
