# Peer review of "Longitudinal Changes of the Ruminal Microbiota in Angus Beef Steers"

_animals, 2022, doi:10.3390/ani12091066_

Round 1

Reviewer 1 Report

Dear authors,

Thank you for your manuscript. I believe that you can improve your manuscript despite not having statistically different results between the time points. This is because you have all sequencing data not deeply analyzed.

As a FAIR principle, your sequencing data must be uploaded to a public database.

Materials and methods

Ln94-96: DNA extraction should be properly described, including conditions for cell lysis.

Ln 105-106, Please describe PCR conditions, taq polymerase used. Were the samples pooled before sequence? If so, how were they normalized?

Ln108 - What were the qPCR conditions?

Did you sequence blanks and negative PCR? Did you use reference standards in your sequencing run?

Ln 119 - Why were you not using CowPi tool? It was designed specifically for this type of data and will give you a more appropriate functional annotation.

Ln 128-132 – Which R packages were used for the statistical analysis? Regarding the A-nova test, is your data normalized?

Results section 3.1

Which microorganisms were driving the clustering of the groups?

Results section 3.2

The results of table 1 can be incorporated in Figures 2 and 3.

The primers used are not covering a good fraction of the archaea community. If the experiment was run with appropriate archaeal primers, you could more appropriately described the community. Therefore, I don’t think it is correct to show results with a maximum of 0.5% archaeal community.

Ln 263-266 should be moved to the results section.

With the current databases, deeper taxonomic information can be presented. However, we should not discuss microbial communities at phylum level. The authors have done a deeper analysis in a similar paper ( https://doi.org/10.3389/fvets.2021.597405).

Discussion

Overall the discussion has to be improved after a deeper analysis of the microbiota taxonomy. There are already studies comparing the microbiota transfer from cows to the calves, there is no need to use pig literature.

Reviewer 2 Report

The present study investigated the longitudinal shift of rumen microbiota in steers from weaning to the end of feedlot raising. The rumen microbial community, both bacteria, and archaea were studied by next-generation sequencing technology.  However, there were no rumen metabolites being analyzed. Instead, rumen metabolic pathways were presented by the sequencing data, the quality of which is a major concern. My questions and comments listed below are for your reference. 

  1. Please provide Information on dietary composition for the cows and steers in the feedlot.
  2. please provide more information about the primer. Is this 16S rRNA gene universal primer since archaea are also targeted? And which hypervariable region?
  3. What is the total number of sequences in each sample? What is the sequencing depth?
  4. When normalized to 5041, how many samples were kept, and how many are removed?
  5. Figure 1:PCA analysis, the percentage of difference explained keeps the number of digits consistent, 29.37%, 5.96%.
  6. Given the percentage of archaea, did you look deeply into which orders of archaea were dominant?
  7. For the bacterial population, I would expect to see results on the genus level abundance.
  8. All the figures, the range of the Y-axis from 0 to 100 as such, and labels need to be adjusted.
  9. Based on the current selection of primer, Whether the archaeal population is underrepresented in the present study, reflect in the discussion section.

Round 2

Reviewer 2 Report

No further comments.